# Learning Unknown from Correlations: Graph Neural Network for Inter-novel-protein Interaction Prediction

## Reproducibility Summary

*This is a report of reproducibility of paper (1), submitted to ML Reproducibility Challenge 2021.*

**Scope of Reproducibility**

In the paper the authors propose a new evaluation that respects inter-novel-protein interactions, and also a new method, that significantly outperforms previous PPI methods, especially under this new evaluation. Therefore we will first inspect if this kind of evaluation is objectively better, and secondly, we will try to reproduce the results of the proposed model in comparison with previous state-of-the-art, PIPR (2).

**Methodology**

For the reproduction we used authors code, slightly changing the pipeline for automatization. We also used PIPR code, where we completely changed the pipeline, to be able to use it on the same datasets as GNN-PPI, but used their function for building the model. The experiments were run on Nvidia Titan X GPU, using around 250 GPU hours altogether.

**Results**

We reproduced the papers results within standard deviations of our repeated experiments. But in some cases, this still means there is a big difference between the performances, which is coming from different train-test splits of the newly proposed splitting schemes. Even with these discrepancies we still managed to (at least partially) confirm all authors claims. The proposed model GNN-PPI performed better than PIPR overall and for inter-novel-protein interactions, evaluation on their proposed schemes predicted the generalization performance better, and their model is also robust for predictions for newly discovered proteins – here our results were surprising, they were even better when the network was built knowing fewer proteins.

**What was easy**

It was easy to run GNN-PPI code on different datasets and with different parameters, as their repository is nicely organized and the code is clearly structured. It was also easy to understand their idea of the problem, the reasons for new evaluation and the framework of their proposed model.

**What was difficult**

In both models used in this reproduction, the environment setup was harder than expected. There was no documentation or comments in the code, which made it hard at first to understand it. Some debugging was needed for GNN-PPI and a lot of code changes for PIPR to train well.

**Communication with original authors**

We communicated with authors through email. They provided some useful clarifications of the method and pipeline.

# 1   Introduction

The study of multi-type Protein-Protein Interaction (PPI) is fundamental for understanding biological processes from a systematic perspective and revealing disease mechanisms. Existing methods suffer from significant performance degradation when tested on different dataset, that was not used for training (in comparison to only dividing one dataset into train and test set). In this paper, authors investigate the problem and find that it is mainly attributed to the poor performance for inter-novel-protein interaction prediction. However, current evaluations overlook the inter-novel-protein interactions, and thus fail to give an instructive assessment.

As a result, they propose to address the problem from both the evaluation and the methodology. Firstly, they design a new evaluation framework that fully respects the inter-novel-protein interactions and gives consistent assessment across datasets. Secondly, they propose a graph neural network based method (GNN-PPI), that uses correlations between proteins for better inter-novel-protein interaction prediction. Experimental results on real-world datasets of different scales demonstrate that GNN-PPI significantly outperforms state-of-the-art PPI prediction methods, especially for the inter-novel-protein interaction prediction.

# 2   Scope of reproducibility

Since authors propose a new evaluation framework, our first task will be to critically inspect their methodology, that is based on different construction of train and test sets. We need to confirm that this kind of dataset splitting is objectively better and mimics the real world problem that they are trying to solve.

Secondly, we will try to reproduce experimental results on real-world datasets of different scales, that were shown in the paper. The authors compare their model to a series of baseline algorithms, some Machine Learning based models (Support Vector Machines, Random Forest and Linear Regression) and some Deep Learning based models (PIPR (2), DNN-PPI (3) and DPPI (4)). Since most of the author claims were made on comparisons of GNN-PPI with PIPR, we will try to reproduce those. The main claims of the original paper, that we will test are the following:

1. GNN-PPI has **higher micro-F1 score** than PIPR on SHS27k, SHS148k and STRING datasets, using *random*, *BFS* or *DFS* testset construction strategies.

2. GNN-PPI predicts **inter-novel-protein interaction** better than PIPR on the same datasets and testset construction strategies as in claim above.

3. Test performance on trainset-homologous testset (trained and tested on same set: SHS27k or SHS148k) under *BFS* and *DFS* partition schemes **reflects the performance of generalizing knowledge** to unseen testset (trained on SHS27k or SHS148k and tested on STRING) better than using *random* partition scheme – micro-F1 scores are more similar, when training and testing GNN-PPI on data split in such way.

4. If we construct the PPI network in GNN-PPI only from the trainset, the micro-F1 score is still better than the one of PIPR. This shows that the trained model is **robust to newly discovered proteins** and their interactions.

# 3   Methodology

For the reproduction we used authors code, slightly changing the pipeline for automatization of multiple runs with different seeds. We also used PIPR code, where we completely changed the pipeline, to be able to use it on the same datasets as GNN-PPI. We only used their function `build_model`, in which we changed last layer activation function from *softmax* to *sigmoid*, because otherwise the model failed in predicting multiple interaction types.

## 3.1   Model descriptions

In our experiments we used the author's model, GNN-PPI and the previous state-of-the-art for multi-type protein interaction prediction, PIPR.

### 3.1.1   GNN-PPI

**GNN-PPI** is a **graph neural network** based method that uses **correlation** between two protein features to predict multiple types of their interaction. Pairwise interaction data are firstly assembled to build the graph, where proteins

serve as the nodes and interactions as the edges. The model is developed by constructing an embedding for each protein to obtain predefined features, then processed by Convolution, Pooling, BiGRU and FC modules (it is called GIN network) to extract protein-independent encoding (PIE) features, which are aggregated by graph convolutions and arrive at protein-graph encoding (PGE) features. Embeddings are pretrained for each amino-acid and combined for the proteins by their amino-acid sequences. The last is Multi-label PPI prediction. For unknown PPIs, we combine their protein feature encoded by the previous process with a dot product, and then use a fully connected layer as classifier for multi-label PPI prediction. For optimization the authors use Adam optimizer.

### 3.1.2 PIPR

**PIPR** employs a **Siamese** architecture of residual **RCNN** encoder to better apprehend and utilize the mutual influence of two sequences. It uses the same pretrained embeddings as GNN-PPI, which are then send through the RCNN, from which we get sequence embedding vectors. This are multiplied with a dot product to form a sequence pair vector. Finally, this sequence pair vector is fed into a multi layer fully connected network with categorical cross-entropy loss function, to predict the multi-label PPI prediction.

### 3.2 Datasets

We trained and tested this two models on three different databases (and their combinations):

(1) **STRING**: this database collected, scored, and integrated most publicly available sources of protein-protein interaction information and built a comprehensive and objective global PPI network, including direct (physical) and indirect (functional) interactions. In this paper, we focus on the multi-type classification of PPI by STRING. It divides PPI into 7 types: reaction, binding, post-translational modifications (ptmod), activation, inhibition, catalysis, and expression. Each pair of interacting proteins contains at least one of them. We use all PPIs of Homo sapiens, which contains 15,335 proteins and 593,397 PPIs.

(2) **SHS27k**: randomly selected 1690 proteins of Homo sapiens subset of STRING, that have 7624 PPIs between them.

(3) **SHS148k**: randomly selected 5189 proteins of Homo sapiens subset of STRING, that have 44488 PPIs between them.

The interactions from the datasets were combined into labels, so that multiple lines that represent different types of interactions between two proteins are combined into one datapoint, where label is a vector of length 7 (number of interaction types) with ones on the indices of interactions that are present and zeros elsewhere.

The authors split datasets so that the test set contained $20\%$ of the interactions. Splitting schemes will be described in the experimental setup more thoroughly, as their evaluation approach was one of their main contributions to the field. Whole datasets are available on the authors GitHub repository, and the combined ones, used for PIPR training can be found on our repository.

### 3.3 Hyperparameters

For GNN-PPI we used the hyperparameters that authors described in Table 10 in the paper. The only change was setting batch size to 2048 when training on STRING dataset, as that was much faster, and the authors also used that in the code on their repository.

For PIPR we did a manual hyperparameter search, to find the parameters where the model works best. We tried different batch sizes, 64, 128, 256, 512, 1024 and 2048; different learning rates, 0.1, 0.01, 0.001 and 0.0001; numbers of epochs, 100, 200 and 300; and RMSprop and Adam optimizers. As the training takes a lot of time, we didn't train the model until the end for all options. With batch sizes, we fully trained it only on smaller datasets and compare the results. We saw that smaller batch size slows the training time a lot, so when using it, we needed to lower the number of epochs. For SHS27k, we saw that it is better to train for 200 epochs with batch size 128, than to do less epochs with batch size 64 (or more with bigger batch size), so we chose this combination. For SHS148k batch size 256 gave better results than 128, and with bigger performance fell. On STRING, the only option was to take batch size 1024, as others were too slow, and with 2048 training also slowed, because we ran out of memory on GPU. Here we only did 100 epochs, because after that, we couldn't use the GPU anymore. We set optimizer to Adam after it proved better on couple runs on the smallest dataset. Learning rate was set just based on first couple epochs (10-20), as we quickly saw that with

0.0001 the training loss hardly even falls, and that with 0.01 and 0.1 it only falls at the beginning, and after couple of epochs, the model stops learning. So we set it to 0.001 for all datasets.

### 3.4 Experimental setup and code

One of the main authors claims is that the usual evaluation scheme (randomly spliting the interactions into train and test set) is not correct from the protein interaction point of view. The result of such split is, that the majority of the proteins were already seen during training, and it is much easier to predict the interactions for such proteins, then for some completely new. To show this, they separate the test set into three subsets: $X_{BS}$ denotes interactions where both proteins were already seen during training, $X_{ES}$ denotes those where one protein was seen, and $X_{NS}$ those interactions, where both proteins are first seen in test phase. In the random testset construction, the $X_{NS}$ set is almost empty (which can be also seen in Table 2), meaning that the testing is not representative for new proteins.

To solve this, authors propose two new strategies, Breath-First Search (BFS) and Depth-First Search (DFS), where the testset is constructed by firstly selecting the root node and then performing the proposed BFS or DFS strategy to select other nodes for the test set (0.2 of the whole dataset in our experiments). As they more thoroughly explained in the paper, these strategies seem to mimic the real world case, where some new cluster of proteins, that tightly interact with each other, is found (BFS) or we just have some new proteins distributed around the previously known network (DFS).

As the authors did in the paper, we then compared the models based on their micro-averaged F1 score (which is the same as the accuracy giving each sample the same importance). We repeated the experiments 5 times where it wasn't too computationally expensive, to get the uncertainty and see how reliable our results are. We were not able to do multiple runs on STRING dataset for each setting, as each run was very expensive, but we believe that the uncertainty here would be very small, as we have a very big dataset. The code for all our experiments is available on this repository.

### 3.5 Computational requirements

The experiments were run on Nvidia Titan X GPU, using around 250 GPU hours altogether. We used different environments for each model – to easily use them we uploaded environment files on our repository.

In Table 1 you can see separate times of training on 300 epochs for GNN-PPI (which was the defeault setting described in the paper) in minutes and prediction on the test set (0.2 of each dataset) in seconds. We can see that training takes quite some time, especially for bigger datasets (18 hours for STRING dataset, but this is not unexpected for a deep learning based approach), but once the model is learned, the prediction is very fast: in range of $10^{-4}$ of a second per one protein pair.

Learning and prediction times are much slower for PIPR model, especially for smaller datasets, where we also took smaller batch size, as it substantially improved the performance. For this reason we only did 200 epochs of training for SHS27k and SHS148k, and it was still much slower than with GNN-PPI. On the STRING set we only did 100 epochs, as we ran out of GPU time we could afford. Long training time also made it harder for us to thoroughly check all hyperparameters and find the best values, as we couldn't possibly do 200 epochs for each parameter choice combination. We can also see that prediction of PIPR is slower, but that is probably because of model using default predict batch size, which is 32 (GNN-PPI uses 256) and could be improved. Even if not, we have around 500 predictions per second, so prediction time shouldn't be a problem.

Table 1: **Training and prediction times.**

|  | Training time [min] | | | Testing time [s] | | |
|---|---|---|---|---|---|---|
|  | SHS27k | SHS148k | STRING | SHS27k | SHS148k | STRING |
| GNN-PPI | 11 | 63 | 1064 | 0.2 | 1.2 | 44 |
| PIPR | 36 | 135 | 602 | 5 | 23 | 291 |

## 4 Results

In this section we will present the results of the experiments that we did to support the claims above. Our results were in some parts far from the authors results, but even so, they support the main claims of the original paper.

## 4.1 Comparison of GNN-PPI and PIPR

Results in this subsection refer to our first two claims. In Table 2 we can see the comparison of the models in question on all three datasets, trained and tested with all three partition schemes. For first two datasets we ran the experiments 5 times, with 5 different random seeds, to see how the performance changes on different sets. We can see that especially for SHS27k the standard deviation for *bfs* and *dfs* splits is very big, which also explain why our results are at some points quite far from the authors. For STRING, multiple runs were too expensive, so we only trained and tested once. But this should be enough to asses the authors claims, because the dataset is so much larger, that the randomness of the split effects the performance less.

The comparison of general performance, that refers to the first claim, can be seen if we look at the micro-F1 score averaged across whole dataset (column *average*). Except for BFS mode on the smallest dataset, where uncertainty is too big, we **can confirm** that in all other cases **GNN-PPI performs better**.

The second claim is that GNN-PPI predicts inter-novel-protein interactions better. The subsets we are observing are denoted with *bs*, standing for both proteins being seen during training, *es*, either of the proteins seen during training and *ns*, neither of the proteins seen before. So for the second claim, we need to compare the performance on subsets *es* and *ns*. The results in the table are **in bold where** uncertainty is small enough that **we can confirm it**. We can also observe that even where it is not, mean is never bigger for PIPR.

Table 2: **Results on test subsets for GNN-PPI (G) and PIPR (P)** on all datasets and splitting schemes. The results show mean F1-micro scores and their standard deviations, based on 5 runs on SHS datasets, and one run on STRING.

| Dataset | Scheme | | Average | Test subsets $X_{BS}$ | $X_{ES}$ | $X_{NS}$ | Test subset sizes $\#BS$ | $\#ES$ | $\#NS$ |
|---|---|---|---|---|---|---|---|---|---|
| SHS27k | rand | $G$ | $\mathbf{0.88} \pm 0.01$ | $\mathbf{0.89} \pm 0.01$ | $\mathbf{0.72} \pm 0.04$ | $0.44 \pm 0.16$ | $1429 \pm 11$ | $91 \pm 10$ | $4 \pm 2$ |
| | | $P$ | $0.81 \pm 0.01$ | $0.82 \pm 0.01$ | $0.6 \pm 0.01$ | $0.25 \pm 0.24$ | | | |
| | BFS | $G$ | $0.54 \pm 0.12$ | / | $0.58 \pm 0.12$ | $0.37 \pm 0.16$ | $0$ | $1180 \pm 103$ | $368 \pm 9$ |
| | | $P$ | $0.49 \pm 0.05$ | | $0.52 \pm 0.05$ | $0.37 \pm 0.08$ | | | |
| | DFS | $G$ | $\mathbf{0.66} \pm 0.12$ | / | $\mathbf{0.68} \pm 0.12$ | $0.5 \pm 0.15$ | $0$ | $1353 \pm 35$ | $191 \pm 25$ |
| | | $P$ | $0.48 \pm 0.06$ | | $0.5 \pm 0.06$ | $0.35 \pm 0.08$ | | | |
| SHS148k | rand | $G$ | $\mathbf{0.92} \pm 0.0$ | $\mathbf{0.92} \pm 0.0$ | $\mathbf{0.75} \pm 0.02$ | $0.19 \pm 0.29$ | $8649 \pm 26$ | $245 \pm 26$ | $3 \pm 3$ |
| | | $P$ | $0.85 \pm 0.0$ | $0.85 \pm 0.0$ | $0.63 \pm 0.03$ | $0.12 \pm 0.21$ | | | |
| | BFS | $G$ | $\mathbf{0.57} \pm 0.05$ | / | $\mathbf{0.59} \pm 0.05$ | $0.39 \pm 0.05$ | $0$ | $7699 \pm 352$ | $1265 \pm 332$ |
| | | $P$ | $0.5 \pm 0.01$ | | $0.51 \pm 0.01$ | $0.37 \pm 0.02$ | | | |
| | DFS | $G$ | $\mathbf{0.79} \pm 0.03$ | / | $\mathbf{0.8} \pm 0.03$ | $\mathbf{0.66} \pm 0.06$ | $0$ | $8247 \pm 82$ | $707 \pm 83$ |
| | | $P$ | $0.54 \pm 0.03$ | | $0.55 \pm 0.03$ | $0.36 \pm 0.07$ | | | |
| STRING | rand | $G$ | $\mathbf{0.95}$ | $\mathbf{0.95}$ | $\mathbf{0.72}$ | $\mathbf{0.99}$ | $118410$ | $268$ | $2$ |
| | | $P$ | $0.87$ | $0.87$ | $0.57$ | $0.33$ | | | |
| | BFS | $G$ | $\mathbf{0.85}$ | / | $\mathbf{0.86}$ | $\mathbf{0.75}$ | $0$ | $107018$ | $11689$ |
| | | $P$ | $0.58$ | | $0.59$ | $0.51$ | | | |
| | DFS | $G$ | $\mathbf{0.89}$ | / | $\mathbf{0.89}$ | $\mathbf{0.83}$ | $0$ | $112328$ | $7058$ |
| | | $P$ | $0.62$ | | $0.63$ | $0.47$ | | | |

## 4.2 Generalization performance

In this subsection we will inspect the third claim, that says that with the newly proposed evaluation protocol, we are better assessing the models generalization abilities. To test that, authors trained both models (we will reproduce results for GNN-PPI) on SHS27k and SHS148k datasets and tested them on the bigger STRING dataset. The test sets of both smaller datasets were now used as validation sets during training, to determine the model from which epoch should be taken as best. The authors then compared these to the results on trainset-homologous testset (here authors don't use validation during training, as they use that set for testing).

The results in Table 3 **confirm claim 3**. As we can see generalizing accuracy severely drops when using random partition scheme ($\sim 0.2$). With newly proposed schemes we get similar, sometimes even better performance when testing on STRING set.

Table 3: **Results of GNN-PPI generalization**.

| Method | Trainset | Testset | Partition Scheme | | |
|---|---|---|---|---|---|
| | | | $random$ | $BFS$ | $DFS$ |
| GNN-PPI | SHS27k-Train | SHS27k-Test | $0.88 \pm 0.01$ | $0.54 \pm 0.12$ | $0.66 \pm 0.12$ |
| | | STRING | $0.62 \pm 0.01$ | $0.56 \pm 0.14$ | $0.61 \pm 0.04$ |
| | SHS148k-Train | SHS148k-Test | $0.92 \pm 0.0$ | $0.57 \pm 0.05$ | $0.79 \pm 0.03$ |
| | | STRING | $0.72 \pm 0.01$ | $0.7 \pm 0.01$ | $0.7 \pm 0.02$ |

### 4.3  Robustness for unknown proteins

In Table 4 we have the comparison of different PPI graph construction methods - *GCA* means that the PPI graph was constructed from all proteins in the dataset, and *GCT* means that PPI graph was constructed only form the proteins in the train set. This results **confirm claim 4**, but in a different way as those in the original paper. It is true that the performance with *GCT* construction method is still better than PIPR method, but contrary to authors results, we get even better performance from the *GCT* construction method than with *GCA*.

Table 4: **Performance of GNN-PPI with different graph construction method**.

| Scheme | Graph | Dataset | |
|---|---|---|---|
| | | $SHS27k$ | $SHS148k$ |
| BFS | $GCA$ | $0.54 \pm 0.12$ | $0.57 \pm 0.05$ |
| | $GCT$ | $0.66 \pm 0.1$ | $0.66 \pm 0.04$ |
| DFS | $GCA$ | $0.66 \pm 0.12$ | $0.79 \pm 0.03$ |
| | $GCT$ | $0.71 \pm 0.09$ | $0.8 \pm 0.02$ |

## 5  Discussion

New evaluation protocol which proves superiority of a newly proposed model could be questionable, because the authors would only select a new evaluation that speaks for their model. But in this case, they also compared the models by random strategy, which is the previously used evaluation method. The new evaluation also has empirical studies to support it and is based on the domain knowledge, so we think that the **authors proposed a good framework** that can be very useful in future research in this field as such performance prediction is more on par with real world situation.

When comparing GNN-PPI and PIPR model, we confirm that **GNN-PPI has better average performance than PIPR**. We should also observe, that the standard deviation for new partiton scheme BFS on the small dataset is too big for us to make certain claims. But we see that on the bigger dataset, it becomes smaller, so we can rely more on this evaluations. There is a bit less certainty when assessing second claim – comparing results in columns $X_{ES}$ and $X_{NS}$ from Table 2. First let's observe that the size of $X_{NS}$ is to small in random partition scheme for all datasets, so we won't compare models on it. Next, we see that with BFS split scheme, training useful models is much harder than for other two. We need a lot more data for models to perform well, and if we have it, then we can claim that GNN-PPI performs better at least when one of the proteins in the pair is seen in training. For both unseen before, we can't claim with enough certainty. So let's say that we **partially proved the second claim** - it holds for random and DFS scheme, and for BFS on bigger datasets. There is one more shortcoming of our approach. We ran out of time to run 200 epochs of training PIPR on STRING dataset. If we had the time, the result could be better, and could potentially beat GNN-PPI. We propose this to be checked in further reproduction.

If we compare absolute scores of GNN-PPI with those that authors described in the paper, they vary quite a bit. This happens because the standard deviations are so big, and the authors might (could be unintentionally) chose the better results of their runs. Their in-depth analysis on test subsets also didn't include uncertainty, which is a problem, as we can see from our results that show that uncertainty is very big and so their results are not reliable.

We **confirmed that the new evaluation protocol is much better in assessing the models generalization abilities**, which was the main problem of the field that the authors were trying to solve. An important thing to observe here is also that when using any model for prediction of PPI, we can use this evaluation protocol and predict its performance in much more detail. We know how many of the proteins that we are interested in are completely new, and how many were already known when training the model, so we can predict the performance separately for any new set, based on performances on test subsets BS, ES and NS.

When comparing the robustness of predictions for unknown proteins, we need to ask ourselves why can we even use the GCA method, that actually uses test data in network construction during training. In most cases that would be a big mistake, but if we look from a protein perspective, this has a practical explanation. We could know which protein exist, what amino-acids they are built of, so we could put them into the network, but we wouldn't know anything about their interactions yet. If we are interested into such proteins, than GCA evaluation will tell us more about our models performance. If we want to know how our model will perform for some newly discovered proteins, we need to inspect evaluation using GCT. In Table 4 we can see a big discrepancy with authors results. They were showing that with GCT the performance falls, which would be expected because we have less information, but in our case, the mean results are better with GCT construction. If we look at the standard deviations, we again see that the differences between results of multiple runs are big, so we can't say with certainty that this results show that GCT is better than GCA. The reason for this could also just be different root node at splitting schemes, and therefore different clusters of unknown proteins. But in any case, it shows that the performance of GNN-PPI with graph construction only from the trainset is much better than the performance of PIPR, which **confirms the fourth claim** and further proves the usefulness of GNN-PPI.

In this reproduction, we left out some parts of the paper that we think should be further verified. We only took PIPR model, and didn't inspect the others for the same tasks. Except for overall performance, also authors didn't compare themselves with other methods, but we believe that for saying that their model is state-of-the-art in inter-novel-protein interaction prediction, they should also inspect other models more thoroughly. With a lot of trouble with setting up PIPR and long training times, we ran out of time for this additional experimentation. We also just superficially grasped the hyperparameter selection for PIPR, and with proper grid search (that would require a lot of computing resources), we could also find better settings for it, or in other case, we could more confidently say that GNN-PPI is better. We also left out the separate results for labels, and ablation study, for which there was no code, but since it could be easily implemented, we suggest it to be done in future. We didn't do it because of time restrictions, and because the authors said in our communications, that it was not the part of original paper, only an addition for requirements of the reviewer, included in arXiv version.

## 5.1 What was easy

The easiest part of this reproduction was to understand the paper and it's ideas. The authors motivate and describe the problem in an easy to understand way, supported by expressive graphics. They support the ideas with domain knowledge, which makes the paper much more insightful. The algorithm for GNN-PPI is clearly described in the paper, so we believe that it would be possible to put it in code without major problems. But their code is also nicely structured and easy to run with various parameters, so we suggest you use it when in need for their method.

## 5.2 What was difficult

For GNN-PPI there was also no environment file, and in the environment descriptions, versions of some libraries were left out, so it took some time to set the environment correctly. You can load the environment file from our repository, to avoid this problem. There was also no documentation of the authors code, so when inspecting the implementation, it was difficult to recognize use of some variables. There was some minor debugging needed before the code ran smoothly. We couldn't use the already trained models for any of our tasks. If we are least judging from their names, they were trained only using the train set, so only useful for checking the accuracy on test part of the same dataset. But this can't be done, because they didn't upload the split information, so we can't build this exact test set. If we wanted to use the models on some other dataset, they would need to upload the model with best validation accuracy, as that one would probably be better.

This were all minor problems compared to the difficulty of running the PIPR model. Their environment file was not useful, and the versions they described on repository were not compatible with each other, which meant that we spent a lot of time finding out which libraries we should load to run their code. The code was not documented and it was

completely unreadable, so it was hard to even know what they were doing. In the end we only used the model building part, and wrote other parts ourselves. Even the building function needed some changes, because they used wrong activation function on the last layer for the model to be able to learn multi-type prediction. Even its description in their paper was not very explanatory, but the reason for that was probably that it wasn't the main part of their research.

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
