# OpenReview forum: "Learning Unknown from Correlations: Graph Neural Network for Inter-novel-protein Interaction Prediction"
_ML_Reproducibility_Challenge/2021/Fall — RC2021_

### Official Review · Reviewer_eEJe · 2022-02-17
**Clear evaluation of novel graph-based model for protein-protein interaction understanding**

**Rating:** 7
**Confidence:** 4

**Review:**

The authors evaluate the reproducibility of GNN-PPI, a graph-based protein-protein interaction paper that proposes a new evaluation method and new modeling of protein-protein interaction.

This reproduction is clearly written. The authors understand the problem and they compare their target model with the valid and recent baseline model. They examine different claims in the original paper and they report running times as well as accuracy results for different parameters. The authors also comment on how hard it was to evaluate different models. Most importantly, they notice that the accuracy of the proposed method they are evaluating still has high variance.

Reproduction authors could have added visualizations or overviews that are not given in the original paper based on experiments they run.

---

### Official Review · Reviewer_RHJg · 2022-02-28
**A comprehensive report with extensive results and inspiring discussions**

**Rating:** 8
**Confidence:** 4

**Review:**

In this report, the authors provided a comprehensive analysis of GNN-PPI experiments as well as the algorithm through reproducing the experiments and their proposed benchmarks. The authors generally agreed on the claims and results in the original published paper. In the meantime, the authors also gave a detailed track of reproducing the original work. I list the pros and cons below. Please correct me if I made anything wrong.

### **Pros**
1. The authors provided a detailed analysis and track on how to reproduce the experiments in the original paper, which may be helpful for other researchers to work on this topic.
2. The listed hyper-params are also helpful for others to reproduce the work.
3. The discussion part, which analyzes the claims from the given results, is very detailed and precise.

### **Cons**
1. The authors can consider providing a more detailed introduction to the original approach and the PPI task. These can be done via following the introduction part as well as Figure 2 in the original paper. I think the report would be even better with an elaborated introduction to the original algorithm.
2. For better readability, the discussion part can be further organized (e.g., via bullets) based on each claim you would like to discuss.

Overall, I think it is an excellent submission to get accepted into this workshop. I would be even better in terms of readability if the authors consider further improving it from the issues mentioned above.

---

### Official Review · Reviewer_mwVX · 2022-03-08

**Rating:** 7
**Confidence:** 4

**Review:**

The project is very detailed. The authors improve the original project with the accompanied code.

---

### Meta-Review · Area_Chair_6Tz8 · 2022-04-07

**Recommendation:** Accept
**Confidence:** 5

**Metareview:**

All reviewers strongly agree on the high quality of reproduction effort in this report, which warrants an acceptance in this venue.

---

### Decision · Program_Chairs · 2022-04-09

**Decision:**

Accept

**Comment:**

Following the recommendation of reviewers and meta-reviewer, the paper is accepted for ML Reproducibility Challenge 2021, and will be published in the upcoming special edition of ReScience Journal.